# Design and Protocol for Beijing Hospital Takayasu Arteritis (BeTA) Biobank

**DOI:** 10.3390/jcm12041516

**Published:** 2023-02-14

**Authors:** Shang Gao, Zhi-Yuan Wu, Yu-Qing Miao, Zhen-Bo Lu, Shu-Ping Tan, Ji-Yang Wang, Cheng-Ran Lu, Zheng-Xi Xu, Peng Li, Yong Lan, Yong-Peng Diao, Zuo-Guan Chen, Yong-Jun Li

**Affiliations:** 1Department of Vascular Surgery, Beijing Hospital, National Center of Gerontology, Institute of Geriatric Medicine, Chinese Academy of Medical Sciences, Beijing 100730, China; 2Chinese Academy of Medical Sciences and Peking Union Medical College, Beijing 100730, China

**Keywords:** takayasu arteritis, biobank, clinical data, sequencing

## Abstract

Background: Although hundreds of studies have been conducted, our understanding of the pathogenesis, indications for surgical intervention, and disease markers of Takayasu arteritis (TAK) are still limited. Collection of biological specimens, clinical data and imaging data will facilitate translational research and clinical studies. In this study, we aim to introduce the design and protocol for the Beijing Hospital Takayasu Arteritis (BeTA) Biobank. Methods: Based in the Department of Vascular Surgery of Beijing Hospital and Beijing Hospital Clinical Biological Sample Management Center, the BeTA Biobank is composed of clinical data and sample data from patients with TAK requiring surgical treatment. All clinical data of participants are collected, including demographic characteristics, laboratory tests, imaging results, operation information, perioperative complications, follow-up data, etc. Both blood samples including plasma, serum and cells, and vascular tissues or perivascular adipose tissue are collected and stored. These samples will promote the establishment of a multiomic database for TAK and help to identify disease markers and to explore potential targets for specific future drugs for TAK.

## 1. Background

Takayasu arteritis (TAK) is a rare vascular disorder that occurs in approximately 40 individuals per million in Asian populations, mainly being prevalent in young women [1,2,3]. TAK presents with chronic granulomatous inflammation in the aorta and its primary branches. Clinically, the early stage is generally characterized by vasculitis, such as limb lameness, skin bruising, hypertension, and pressure difference in the upper limbs, and serious clinical progression and complications such as coronary ischemia, renal artery stenosis, aortic reflux and aneurysms may occur in the late stage [4,5,6]. Patients with TAK are mainly treated with general immunotherapy drugs such as glucocorticoids, immunosuppressants and biological agents, but no specific drugs have been introduced to treat TAK. Approximately 30% of patients still require surgical interventions including open surgery and endovascular repair [7,8]. The ten-year long-term survival in these patients is about 80–90%, but primary patency is lower than 50% [9]. What happens inside the culprit vessels has attracted researchers during the past decades.

Hundreds of clinical and translational studies have been conducted to explore the biomarkers, genetic variants, pathophysiological mechanisms, and even the ultimate aims of effective drugs for TAK. General inflammation markers including ESR (Erythrocyte sedimentation rate) and CRP (C reactive protein) were adopted as indicators for TAK activity, progression, or relapse [10,11]. Other circulating markers such as MMP-2 and IL-6 are promising markers, but are still far from usable in clinical practice [11,12]. Imaging markers in PET-CT could serve as an optimal marker for monitoring disease activities, but features of this marker limit its use population [13].

The current brief theory of the pathophysiology of TAK is that unknown stimuli cause dendritic cells to produce HLA molecules, release IL-18 and chemokines to activate T cells, and further activate B cells to produce autoantibodies, leading to apoptosis of vascular endothelial cells. Activated Th17, CD4^+^, and CD8^+^ T cells produce IFN-γ, IL-6 and TNF-α, further activating macrophages and giant cells. These cells then release reactive oxygen species IL-23, IL-6, IL-1, IL-12, TGF-β1, MMPs, FGF-2, PDGF, and VEGF [14,15,16], resulting in successive biological vascular oxidative damage, extracellular matrix degradation, elastic layer destruction, intimal hyperplasia and neoangiogenesis, and vascular remodeling [4,17,18,19,20,21]. However, due to the lack of culprit vessel specimens and the complexity of the disease, our understanding of the pathophysiology of TAK remains largely insufficient.

In terms of genetic load, studies have only explored the common mechanisms of large-vessel vasculitis (LVV) represented by TAK and Giant cell arteritis (GCA) through cross-phenotypic analysis in Caucasian populations. Ultimately, HLA-DRA, HLA-DRB1 (GCA-related), HLA-B, MICA (TAK-related) in the HLA region, and IL12B and KDM4C outside the HLA region were determined to be specifically associated with LVV [22,23]. Other studies have shown that FCGR2A/FCGR3A is the genetic susceptibility site for TAK [24]. However, there have been no studies performing more in-depth genetic locus association analysis in Asian populations, which are more prone to systemic vasculitis.

To fill these gaps, it is imperative to integrate clinical data with biological samples and imaging markers. Therefore, we have established a biobank with these multiple-dimension datasets in our vascular center. We named this biobank the “Beijing Hospital Takayasu Arteritis Biobank”, or “BeTA”. This multiple-dimension biobank aims to facilitate translational research and clinical studies on TAK. One of its features is that enrolled patients are mainly those requiring surgical interventions, covering both active and inactive phases according to the Kerr score (developed by NIH) and ITAS 2010 (developed by Vasculitis Group of Indian Rheumatology Society). 2022 ACR/EULAR TAK classification criteria will also be considered. Patients who have suffered re-interventions because of re-stenosis or graft occlusion will also be admitted into this project. Another focus of the biobank is to provide unprecedented opportunities for tissue sequencing with state-of-the-art techniques like bulk RNA-seq [25], single-cell RNA sequencing [26] and multiomics. At the same time, using the sequencing data, we can also find more genes and polymorphisms related to TAK in Asian populations and compare them with Caucasian populations to find the genetic load of populations with different genetic backgrounds. The collection of clinical data, biological specimens, and imaging data from the center will boost studies of patients with TAK in a more comprehensive manner. This multiple-dimension biobank is the first biobank specifically focusing on TAK. Future studies using these data will be based on the scientific objectives of this biobank, as follows: (1) to identify indications and optimal timing for surgical intervention in TAK; (2) to perform in-depth studies of the natural course and mechanism of TAK on the molecular, cellular, tissue and other levels; (3) to identify biomarkers that can reflect the prognosis and activity of TAK; and (4) to explore targets for drug treatment.

## 2. Method

### 2.1. Database Plan

The BeTA Biobank is a single-center biobank characterized by a clinical database and imaging data for TAK patients in China, initiated and implemented by vascular surgeons in Beijing Hospital based on research topics related to surgical intervention strategies. The BeTA Biobank aims to collect vascular tissue samples, peripheral blood and other relevant clinical data, and imaging data for scientific research and analysis.

### 2.2. Ethical Review

The investigation of human samples here conforms with the principles of the Declaration of Helsinki. Based on the research topic of surgical intervention strategies for TAK, the Medical Ethics Committee of Beijing Hospital has approved the establishment of the BeTA biobank and the performance of relevant basic research based on the database and has determined that surgical interventions for TAK are within the scope of medical ethics. PThe registration number is ChiCTR1800018752.

### 2.3. Venue and Population

The BeTA Biobank is based on and relies on the Beijing Hospital Biobank, which uses the internationally recognized RuRo sample bank management software. This tool can effectively and accurately track the complete process of sample collection, storage and output in an intelligent manner. The center covers an area of thousands of square meters and has a professional laboratory and sample preservation base. A gas-phase nitrogen storage system is used inside the base, and a −80 °C turnover refrigerator is placed in the operating room to support the cryopreservation of blood and tissue samples from Takayasu arteritis patients. Per national regulations, the biobank center has obtained an administrative license for the preservation of human genetic resources and has been approved by the ethics committee of the Chinese Medical Association to engage in preservation activities.

### 2.4. Recruitment of Groups

According to the relevant requirements of the BeTA Biobank, the recruitable population meet the following inclusion and exclusion criteria (Table 1).

### 2.5. Recruitment Procedure

Vascular surgeons from the Department of Vascular Surgery in Beijing Hospital recruit eligible patients in inpatient or outpatient clinics. Patients who consent to participate sign a written informed consent form during the visit to the inpatient or outpatient clinic. When a patient requires emergency surgery due to worsening symptoms of TAK, written consent of the patient or family is required. Verbal consent must subsequently be confirmed in writing by the patient or their family. If written informed consent cannot be obtained, the relevant biological samples will not be stored.

### 2.6. Collection and Processing of Biological Materials

We routinely collect blood on the second day of hospitalization along with other blood tests. Two tubes of EDTA anticoagulation (plasma + white membrane layer + erythrocytes) and conventional whole blood (serum + blood clots) are conventionally adopted for collection. After assigning a unique identification code, the sample is sealed in the biospecimen center of the Beijing Hospital Biobank. Blood samples from donors are also collected to serve as controls. After collection, the blood sample is temporarily stored at 2–8 °C and processed within 24 h. Serum, plasma, white membrane layer including leukocyte and erythrocytes are centrifuged from EDTA anticoagulant tubes and stored below −80 °C in several cryopreservation tubes (500 μL/tube). Subsequent freeze-thaw can be used for genetic testing and live-cell extraction. Routine whole blood samples can be used for the detection of routine physiological and biochemical indicators.

For patients with TAK who meet surgical indications, we routinely prepare several cryopreservation tubes and formaldehyde sample tubes and collect tissue samples of vascular tissue or perivascular adipose tissue in the BeTA Biobank. Specifically, the operator prepares sample preservation tubes before collection and labels the tubes with a unique identification barcode. Subsequently, careful saline cleaning and washing is applied on the operating table and the samples are divided into two groups: liquid nitrogen cryopreservation with or without RNAlater, and formaldehyde fixation (for pathological diagnosis or scientific research). The process of dispensing and marking after leaving the body does not exceed 30 min. Depending on the surgeries performed in the patients with TAK, the location of vascular tissues or perivascular adipose tissue ranges from the carotid artery to the renal artery, from the ascending aorta to the abdominal aorta, or even other visceral arteries. Furthermore, the thrombus in the graft is also collected. These samples are stored in the biological sample bank on the same day (Table 2, Figure 1).

### 2.7. Clinical Information Collection

Basic clinical information and related follow-up data for patients mainly come from the export of medical record system data, and some of the information comes from direct inquiries to patients. The follow-up information is mainly derived from regular follow-up by professional follow-up personnel and the data collected during the review process. We also use account passwords to manage the clinical databases to ensure that the clinical information collected in the database can only be used by database researchers for scientific research (Table 3).

### 2.8. Statistical Analysis Methods

Under the guidance of statistical experts, SPSS 26.0 (IBM, New York, US) software is used for statistical analysis. For normally distributed data, the measurement is calculated as a mean ± standard deviation. The Student’s t-test is used for the comparison between the groups, the chi-square test or the Fisher exact probability method is used for the counting data, and the odds ratio is used to express the relative risk. Nonnormal distribution data are expressed as median and quartiles, and a nonparametric test is proposed for data that needs statistical calculation. The chi-square test analyzes the correlation between influencing factors such as perioperative immunotherapy and disease outcome. Logistics regression is analyzed for risk factors for developing bridging vessels restenosis and complications. The significance level is α = 0.05, and *p* < 0.05 is statistically significant.

### 2.9. Patient and Public Participation

The database was established by vascular surgeons, statistical experts and biobank biospecimen managers. No patients or the public commented on the study design, and patients were not consulted to formulate patient-relevant results.

## Figures and Tables

**Figure 1 jcm-12-01516-f001:**
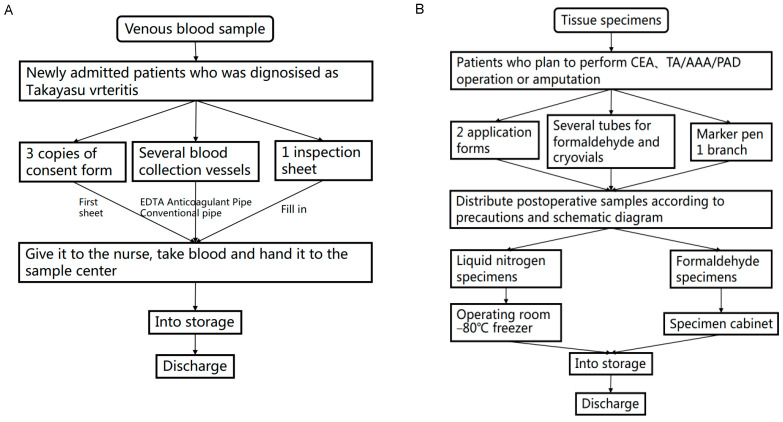
Collection process for Takayasu arteritis specimens. (**A**) Collection process for venous blood specimens of Takayasu arteritis. (**B**) Collection process for vascular wall tissue specimens of Takayasu arteritis.

**Table 1 jcm-12-01516-t001:** Inclusion and exclusion criteria for patients with Takayasu arteritis.

Enrollment Criteria	Exclusion Criteria
The diagnosis of Takayasu arteritis conforms to the 2022 ACR/EULAR TAK classification criteria	Patients with other systemic autoimmune diseases
Persistent active disease remains under immunosuppressive therapy	Patients with severe hematological and endocrine system lesions or tumors
Conformance indications for surgical intervention: 1. Imaging examination shows severe stenosis (> 70%) or occlusion of the aorta and its branches 2. Intractable renovascular or aortic constricting hypertension 3. Ischemia of heart, brain, limbs and other organs 4. Aortic aneurysm-like lesions	Patients with severe heart, lung, liver, kidney and other important organ lesions, unable to tolerate anesthesia and surgery
	Patients with acute or chronic infectious diseases (except tuberculosis)
	Patients with mental disorders or who are unable or unwilling to cooperate with treatment for other reasons
	Patients who are pregnant before surgical intervention

**Table 2 jcm-12-01516-t002:** Biological sample collection.

Biomaterials	Acquisition Time	Saved State	Position
Peripheral blood: EDTA anticoagulation	The morning after admission	−80 °C RNAlater	Peripheral venous blood
Peripheral blood: routine biochemical serum	The morning after admission	4 °C preservation	Peripheral venous blood
Samples of the vessel wall of Takayasu arteritis	Intraoperative dispensing	Formalin specimen, encapsulated at normal temperature	Diseased vascular segments
Samples of the vessel wall of Takayasu arteritis	Intraoperative dispensing	−80 °C RNAlater	Diseased vascular segments

**Table 3 jcm-12-01516-t003:** Timeline for clinical follow-up information collection for Takayasu arteritis.

Follow-Up Programs		Enlistment	1 Month	3 Months	6 Months	9 Months	12 Months
Medical history	Symptom	√	√	√	√	√	√
	Sign	√	√	√	√	√	√
Blood	ESR	√	√		√		√
	C-reactive protein	√	√		√		√
Image	Ultrasound of the target vessel	√	√	√		√	
	Target vascular CTA	√			√		√
Medication	Immunotherapy	√	√	√	√	√	√
	Antihypertensive therapy	√	√	√	√	√	√
	Antithrombotic therapy	√	√	√	√	√	√
Quality of life	Preoperatively SF-36	√					

## Data Availability

There is currently no online data available for download.

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
