# Peer review of "Design and Protocol for Beijing Hospital Takayasu Arteritis (BeTA) Biobank"

_jcm, 2023, doi:10.3390/jcm12041516_

Round 1
Reviewer 1 Report
This manuscript is concise, well organized, and potentially acceptable as a protocol. A few suggestions are proposed:
1. Show the citation points of Table 2 and Figure 2.
2. Lines 135-136: Pathological diagnosis and formaldehyde fixation should be the same.
Author Response
Thank you for taking the time to review our manuscript
For question 1, Table 2 is a summary of the main points of our vascular surgery center's methods for treating arteritis tissue or blood, not from other documents. Figure 2 is a pattern diagram. We have completed the image processing work by ourselves, and this picture is not from professional literature, treatises or books, so there is no reference.
For question 2, thank you for your correction. I'm very sorry that the original manuscript has not explained this aspect in detail and has been revised in the original manuscript. The formaldehyde fixation group is divided into two parts, one part is delivered to the pathology department for surgical pathology diagnosis, and the other part is left in the biobank for further exploration in scientific research direction, such as immunofluorescence.
Reviewer 2 Report
This is a terrific biobank of samples and data from TAK patients. Certainly, it will result in plenty of novel findings on the pathophysiology of the disease. The manuscript is well-written and it describes the biobank protocol in a comprehensive way. I have a few suggestions for the manuscript.
Comment: the use of the NIH criteria to assess disease activity in TAK may be a problem as this tool lacks sensitivity. It may be necessary to adapt disease activity assessment to novel tools currently under development.
Table 1 – I suggest to change the 1990 ACR classification criteria to the 2022 ACR/EULAR classification criteria for TAK.
Table 1 – Please change “Patients with other rheumatic immune diseases” to “other systemic autoimmune diseases”. I guess the authors would exclude patients with Crohn’s disease, multiple sclerosis…
Page 3 (lines 114 to 115) – In the case of emergency surgery, in which the patient cannot provide written consent, why do you not get written consent from a relative rather than verbal consent?
Author Response
Thank you for taking the time to review our manuscript
For comment 1 and 2, the 2022 ACR/EULAR TAK classification standard has not been adopted by our center at the time of the draft of Table 1. We tried to use this standard in the follow-up diagnosis and treatment, and the diagnosis result is better than the 1990 version, so we will also use the 2022 ACR/EULAR TAK classification criteria, and for the activity standard. We will also add the ITAS2010 standard as one of the identification methods of the activity standard in the future, the 2022 ACR/EULAR TAK classification criteria will also be considered. Thank you for your correction.
For comment 3, thank you for your correction. We have corrected this question.
For comment 4, I'm sorry for our omission. We rarely encounter emergency surgery in the operation of Takayasu arteritis in our center. Therefore, our informed consent policy on emergency operation of Takayasu arteritis is not perfect. We will obtain the written consent of the family first according to your suggestion. When the written consent is also difficult to obtain, we will use oral consent.